# Scale-invariant magnetic textures in the strongly correlated oxide NdNiO$_3$

Jiarui Li [1], Jonathan Pelliciari[1], Claudio Mazzoli[2], Sara Catalano[3,8], Forrest Simmons[4], Jerzy T. Sadowski [5], Abraham Levitan[1], Marta Gibert[6], Erica Carlson[4,7], Jean-Marc Triscone[3], Stuart Wilkins [2] & Riccardo Comin [1]*

Strongly correlated quantum solids are characterized by an inherently granular electronic fabric, with spatial patterns that can span multiple length scales in proximity to a critical point. Here, we use a resonant magnetic X-ray scattering nanoprobe with sub-100 nm spatial resolution to directly visualize the texture of antiferromagnetic domains in NdNiO$_3$. Surprisingly, our measurements reveal a highly textured magnetic fabric, which we show to be robust and nonvolatile even after thermal erasure across its ordering temperature. The scale-free distribution of antiferromagnetic domains and its non-integral dimensionality point to a hitherto-unobserved magnetic fractal geometry in this system. These scale-invariant textures directly reflect the continuous nature of the magnetic transition and the proximity of this system to a critical point. The present study not only exposes the near-critical behavior in rare earth nickelates but also underscores the potential for X-ray scattering nanoprobes to image the multiscale signatures of criticality near a critical point.

[1] Department of Physics, Massachusetts Institute of Technology, Cambridge, MA 02139, USA. [2] National Synchrotron Light Source II, Brookhaven National Laboratory, Upton, NY 11973, USA. [3] DQMP, University of Geneva, 24 quai Ernest-Ansermet, 1211 Genève 4, Genève, Switzerland. [4] Department of Physics and Astronomy, Purdue University, West Lafayette, IN 47907, USA. [5] Center for Functional Nanomaterials, Brookhaven National Laboratory, Upton, NY 11973, USA. [6] Physik-Institut, University of Zurich, Winterthurerstrasse 190, 8057 Zürich, Switzerland. [7] Purdue Quantum Science and Engineering Institute, Purdue University, West Lafayette, IN 47907, USA. [8] Present address: CIC Nanogune, Tolosa Hiribidea 76, 20008 Donostia, Spain. *email: rcomin@mit.edu

Transition metal oxides (TMOs) are characterized by rich phase diagrams where different collective electron phases (metal, insulator, superconductor) cohabit in proximity to one another[1–3]. When these electronic phases compete, they segregate at nano- or mesoscopic length scales, giving rise to complex, multiscale domain textures[4–6]. Nowadays, there is overwhelming evidence of domain pattern formation for an array of emergent phenomena, including metal-insulator transitions[7–11], charge/magnetic ordering[12–14], and unconventional superconductivity[15–17]. Several studies have identified these spatial textures as essential underpinnings for phenomena in TMOs and other correlated electron systems[5,13,18–21].

Rare earth nickelates are a fertile platform to realize and explore the emergent physics brewing in proximity to electronic symmetry-breaking phenomena, thanks to the cooperative action of the coupled charge, spin, and lattice degrees of freedom. In this family of compounds, the dominant structural/electronic instability sets the stage for a first-order transition (P2$_1$/n to Pbnm) to a bond-disproportionated phase, which is accompanied by a concurrent insulator-to-metal transition (IMT). The low-temperature electronic phase then undergoes a second transition to an antiferromagnetic (AFM) ground state[22–24]. The nature of the AFM transition ostensibly changes from second-order for smaller cations (Sm, Eu, Y) to first-order for larger ones (Nd and Pr)[22,23]. This abrupt change in the nature of the magnetic transition has been a long-standing puzzle, one that has been obfuscated by the dominance of the structural/electronic transition and by the imprinting of its hysteretic signatures onto the secondary magnetic order parameter. For this latter reason, a conventional temperature scaling analysis of the AFM transition is not applicable for NdNiO$_3$ and PrNiO$_3$, hindering a full elucidation of the magnetic transition. Whether the nature and universality class of the magnetic transition truly changes with the rare earth cation size across the bifurcation point, and whether the IMT and AFM transition lines meet at a tricritical point, are only a few of the outstanding questions that have hindered a full understanding of the electronic phase diagram of rare earth nickelates.

Recent studies have aimed at these scientific questions by examining the spatial organization of electronically ordered domains. This intense activity has exposed a remarkably textured nanoscale fabric manifested by the coexistence of electronic (metal/insulator) nanoregions across the IMT[7,11]. Even more recently, nanoscale evidence has been reported of a continuous temperature evolution of conductive one-dimensional domain walls in NdNiO$_3$, despite this system nominally undergoes a first-order transition[10]. These observations reignited the debate on the nature of the phase transition in nickelates and on whether the charge and spin degrees of freedom are coupled or decoupled across the IMT. These matters have remained pending owing to the lack of direct nanoscale visualization of the AFM domains in these systems, and have here motivated the development of experimental tools to provide a comprehensive view of these phenomena in inherently phase-separated materials.

In the present work, we examined the spatial organization of AFM domains in an epitaxial bulk-like (26 nm) NdNiO$_3$ thin film grown on NdGaO$_3$ (110)$_{ortho}$ substrate with $T_{IMT} = T_{Néel}$ ~ 180 K (Supplementary Fig. 1). Our experimental strategy relies on phase transition theory that second-order transitions are characterized by critical behavior whereby various physical quantities exhibit power-law scaling. Criticality generically implies power-law correlations, which are dynamic in time. However, when the critical point involves random field disorder[25], barriers to equilibration grow exponentially with proximity to the critical point[26], and the power-law distribution becomes frozen in time, manifesting itself as a static, fractal domain structure[27]. Recently, nanoscale probes have been developed and applied to visualize scale-invariant

electronic domain patterns in various correlated systems, directly exposing their proximity to critical behavior[12,21,28]. At the same time, AFM domain patterns have evaded conventional magnetic imaging techniques, including scanning probes of the local magnetization, which are insensitive to antiferromagnetism[29,30], or electron microscopy techniques, which are exquisitely sensitive to surface phenomena[31–34]. To date, a bulk-sensitive, submicron-level nanoprobe of AFM domain textures has been lacking.

## Results

**Scanning resonant magnetic X-ray scattering nanoprobe.** For this study, we combined resonant magnetic soft X-ray scattering (RMXS) with special X-ray focusing optics (Fresnel zone plate) to visualize AFM order at the 100 nm scale[35]. RMXS has been extensively used to probe magnetic ordering in TMOs, especially in thin film materials that fall below the detection limit of neutron scattering[36]. When the incident photon energy is tuned to the Ni 2p-3d core level transition (852.5 eV), the atomic scattering amplitudes become spin-dependent, enabling detection of periodic AFM ordering of the Ni spins[37–40]. However, conventional RMXS measurements have been historically performed with macroscopic X-ray beam footprints ($\gtrsim$ 100 microns), which average out over spatial inhomogeneities in the magnetic domain distribution. Here, we used this scanning RMXS nanoprobe (hereafter, nano-RMXS) to visualize symmetry-breaking electronic orders in reciprocal and real space simultaneously, in a unique combination of scattering and microscopy. The X-ray probing depth at resonance ($\gtrsim$ 100 nm) guarantees access to bulk AFM textures.

The nano-RMXS setup is illustrated in Fig. 1. The macroscopic X-ray beam is focused by the zone plate lens, which produces a series of focal spots along the optical axis[41]. An order sorting aperture is used to propagate only the first-order beam and filter out all other diffraction orders. During the entire experiment, soft X-rays are focused on the sample surface with a lateral beam waist of ~ 100 nm (Fig. 1a). The NdNiO$_3$ sample and the charge-coupled detector are positioned to collect photons scattered at the AFM ordering vector $\mathbf{Q}_{AFM} = [¼, ¼, ¼]_{pc}$ (in reciprocal lattice units and pseudocubic notation), as shown in Fig. 1b[37–40]. The AFM domain maps are obtained upon scanning the X-ray beam by a synchronized translation of the focusing optics to within a ~5 nm accuracy. The integrated scattering intensity is subsequently extracted at each beam position, yielding the AFM domain maps shown in Fig. 1c. The magnetic scattering intensity, which is proportional to the squared amplitude of the AFM order parameter, exhibits multiscale inhomogeneities even at a temperature (130 K) far below the ordering transition. The AFM domains tend to coalesce into interconnected submicron clusters but, at the same time, smaller spatial structures can be observed, reflecting a broad distribution of length scales.

It is crucial to understand the origin of this apparent inhomogeneity in the magnetic domain distribution. One possible scenario is that we are observing the phase separation of AFM-insulating and paramagnetic-metallic domains that persists below $T_{IMT}$[7]. To elucidate the electronic texture of the "weak AFM" region where the magnetic scattering intensity is low, we imaged the X-ray absorption spectral signature at the Ni-$L_3$ resonance using X-ray photoemission electron microscopy (XPEEM), where the Ni valence state properties can be visualized down to 50 nm spatial resolution. At 100 K, far below $T_{IMT}$ ($T_{Néel}$), the XPEEM map provides direct evidence that there are no metallic domains within the whole field of view as shown in Supplementary Fig. 3c. Therefore, the textured AFM domain landscape we observe is not owing to the coexistence of metal and insulator domains.

The presence of AFM spatial textures in a homogeneous insulating state can be understood to originate from a distribution

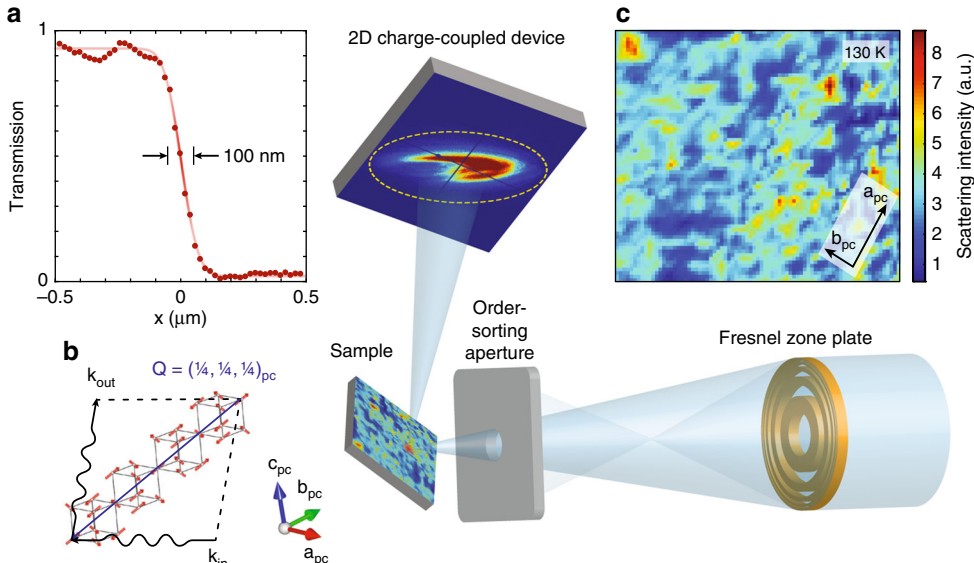

**Fig. 1** Experimental setup and nanoscale antiferromagnetic landscape in NdNiO$_3$. The combined use of a Fresnel zone plate and an order sorting aperture delivers a ~ 100 nm focused soft X-ray spot at the sample. The magnetic Bragg diffraction patterns are visualized with a fast area detector. **a** Knife edge scan, demonstrating a spot size of around 100 nm at 852.5 eV (Ni-$L_3$ resonance). **b** The sample was oriented to intercept the antiferromagnetic diffraction peak at $\mathbf{Q}_{AFM} = [\frac{1}{4}, \frac{1}{4}, \frac{1}{4}]_{pc}$ (reciprocal lattice units in pseudocubic notation) arising from ordering within the Ni spin sublattice[38,39]. **c** The spatial map of magnetic scattering intensity exposes a highly inhomogeneous electronic landscape at the nanoscale. Arrows indicate the local pseudocubic crystallographic orientation. Each arrow length spans 1 μm in the sample frame of reference. Notice that in this scattering geometry, owing to the sample surface not being normal to the incident X-ray beam, additional geometric factors need to be accounted for when converting lateral beam translation footprint (defined in lab frame) onto the sample surface (sample frame)

of different magnetic domains. As the ordering vector $\mathbf{Q}_{AFM} = (\frac{1}{4}, \frac{1}{4}, \frac{1}{4})_{pc}$ has eight equivalent direction in the pseudocubic cell, the spiral magnetic configuration shown in Fig. 1b lifts the eightfold degeneracy in such a way that a specific magnetic domain only generates scattering intensity along two out of eight directions. The structure factor of the spin spiral magnetic structure along $(1/4, 1/4, 1/4)$ only become nonzero when measuring at $\mathbf{Q}_1 = (\frac{1}{4}, \frac{1}{4}, \frac{1}{4})_{pc}$ and $\mathbf{Q}_2 = (-\frac{1}{4}, -\frac{1}{4}, -\frac{1}{4})_{pc}$ and is zero for the other ordering vectors. With only two out of eight equivalent $\mathbf{Q}_{AFM}$ ordering direction can be accessed in the current geometry, we effectively probe a fraction of $2/8 = 25\%$ of magnetic patches, assuming an equal population of the symmetry-equivalent AFM Q-domains.

**Memory effect**. To investigate the evolution of the AFM domain patterns across the Néel transition, the sample was thermally cycled from 130 K to 210 K (above $T_{Néel}$) and back. Figure 2 displays the nanoscale AFM order parameter within the same field of view before the hysteresis region (Fig. 2a), within the Néel transition (Fig. 2b), and after completion of a full hysteresis cycle back at 130 K (Fig. 2c). The corresponding temperatures of the maps in Fig. 2a–c are marked onto the intensity-temperature curve in Fig. 2d. The scattering intensity decreases during the warming cycle, and no scattering intensity is detected above the transition even with a macroscopic X-ray beam (200 μm), indicating a complete erasure of AFM order. Interestingly, the domain distribution remains largely similar between 130 K and 180 K, despite a substantial metal/insulator phase separation being expected near $T_{IMT} \sim 180$ K[7].

As the sample is cooled back down to 130 K, the full thermal cycle does not appear to reset the domain morphology, as hinted by the overlaid domain contours of Fig. 2a (130 K) onto the other maps. This observation reflects a reminiscence of the spin degrees of freedom even after full quenching above $T_{Néel}$, suggesting a sizeable memory effect during the reentrance into the AFM phase

(for a more-quantitative analysis of this memory effect, see Supplementary Fig. 4 and supplementary discussion). This apparent resilience against temperature variations suggests the occurrence of domain pinning which is robust across the magnetic ordering transition. This memory effect, which to the best of our knowledge has never been studied at the nanoscale across a Néel transition, has also been observed in other strongly correlated electron systems[42].

The temperature-dependent spatial maps of AFM domains offer further insights onto the origin of the hysteretic, first-order-like behavior of the magnetic scattering intensity, as apparent from Fig. 2d[38,43]. The observed sixfold suppression of the global order parameter between 130 K and 180 K (Fig. 2d) is found to arise not out of shrinking magnetic domains, but rather from a reduction in the order parameter within each domain (see Supplementary Fig. 6 and corresponding supplementary discussion for additional details on the analysis). This local reduction is of the same order as that of the spatially averaged order parameter—suggesting that, although the domain structure remains largely unchanged, the order parameter is uniformly suppressed across the transition. In other words, the magnetic order parameter develops via a continuous transition inside the static AFM domains, and not by means of a nucleation and growth mechanism. The latter are, in turn, constrained within the insulating domains which themselves behave hysteretically (following a first-order metal-insulator transition). We note that the continuous evolution of the AFM order parameter in the interior of the domains cannot be directly compared with similar findings of smoothly evolving metallicity in 1D conducting channels as visualized in ref. [10], as the latter were proposed to represent regions where antiferromagnetism is locally suppressed.

**Scale-invariant domains textures**. The availability of nanoscale spatial maps of the AFM order parameter also allows for higher-level statistical analysis of the anatomy of the domain

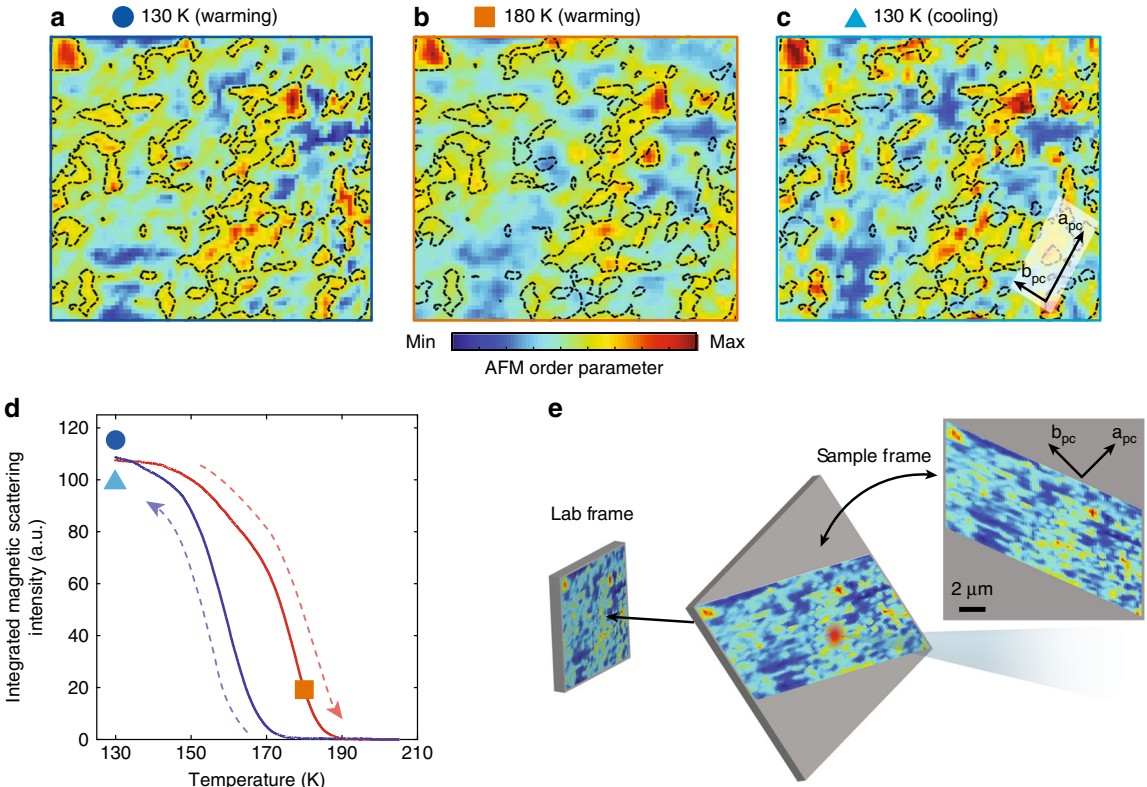

**Fig. 2** Magnetic texture vs. temperature. **a–c** Evolution of the spatial AFM order parameter (square root of scattering intensity) during a thermal cycle across $T_{\text{Néel}}$ at temperatures corresponding to the plot markers in (**d**). Min/max color scales are specific to each map. Black dashed contours outline the AFM domains at 130 K (warming cycle (**a**) and are also overlaid in (**b**) and (**c**) to highlight the evolution. The $a_{pc}$ and $b_{pc}$ crystallographic axes are indicated in (**c**); the arrow length corresponds to 1 µm in the sample frame. **d** Temperature dependence of the $[\frac{1}{4}, \frac{1}{4}, \frac{1}{4}]_{pc}$ AFM Bragg peak intensity, measured using a macroscopic X-ray beam, shows a typical first-order like hysteresis [red (blue) curve is the warming (cooling) cycle]. The temperatures of maps **a–c** are marked out as well. **e** The projected AFM landscape map on the sample surface. Note that the measured sample area is not square, and the projected shape, size, and orientation of the AFM domain landscape are shown in the top right inset

distribution. In Fig. 3a, we binarized the 130 K scattering map (warming cycle) using an intensity threshold point designed to select a coverage fraction of 25% of the total area. Correspondingly, we separate the magnetic domains (yellow, 25% of the total probed area) probed in the current geometry from all other domains (blue, the other 75% of the total probed area). From the binarized maps (Fig. 3a), various descriptors of the domain morphology can be inferred via statistical analysis. Figure 3b shows the domain area distribution histogram. We employ logarithmic binning, a standard technique for power-law distribution analysis[44]. Most of the points fall onto a single line for the three maps, without the use of any cross-normalization. Quantitatively, these distribution functions exhibit power-law scaling spanning two decades in the domain area with an average critical exponent $\tau = 1.25$. A similar form of power-law scaling behavior in the charge degrees of freedom has been previously observed in strongly correlated oxides with various techniques[12,21,27]. Figure 3c, summarizes the relationship between the perimeter (area) and the gyration radius for all domains. The domain perimeter ($P$) and area ($A$) scale as a power of the gyration radius $R_g$ ($P = R_g^{d_h}$, $A = R_g^{d_v}$) across two decades of scaling. The gyration radius of an individual domain is defined as $R_g = \sqrt{\overline{(r - \bar{r})^2}}$. The fitting analysis returns the averaged exponents $d_h = 1.23$ and $d_v = 1.78$, known as the hull and volume fractal dimensions[45]. These fractal dimensions cannot be trivially explained by an uncorrelated (i.e., non-interacting) percolation model, in which local quenched disorder is the sole determiner of the local magnetic order parameter at each spot in space. The fact

that the critical exponents of uncorrelated percolation ($d_h = 7/4 = 1.75$, $d_v = 91/48 = 1.8958$, and $\eta = 5/24 = 0.208$[45]) are inconsistent with our data-extracted exponents (Fig. 3e and Supplementary Table 1) is an indication that Coulomb interactions play a crucial role in the pattern formation. Furthermore, our 25% single $\mathbf{Q}_{\text{AFM}}$ domain coverage is far from the *2d* critical uncorrelated percolation threshold on a square lattice $p_c = 0.59$. Therefore, the scale-free distribution of the AFM domains in our NdNiO$_3$ thin film cannot be attributed to 2d uncorrelated percolation effect[45]. Domains either touching the boundary of the maps or with the size close to single pixel size are excluded from the analysis (hollow markers). The smallest domains are subjected to non-universal, short-distance physics, whereas spanning clusters are left out as their dimension can be affected by the finite spatial field of view. Most of the excluded points still fall near the scaling curve, possibly suggesting a larger scaling range that could be accessed with higher resolution or larger fields of view. Details of the fitting results for each map are described in the Supplementary Table 1. These observations of non-integral exponents and scale-free distribution altogether point to an unexpected fractal nature of AFM domains in NdNiO$_3$[8,21,27,31,46].

The formation of static fractal electronic patterns hints at nearby criticality. By tuning the temperature, we can locate the system in the vicinity of the critical point. In Fig. 3e, we extracted the pair connectivity function $g_{\text{conn}}(r)$ for each map. The pair connectivity $g_{\text{conn}}(r)$ is defined as the probability that two sites separated by a distance $r$ belong to the same connected finite cluster[45,47]. The pair connectivity function was fitted by a power-law function with an exponential cutoff $r^{-n}\exp(-r/\xi)$ where $\xi$ is

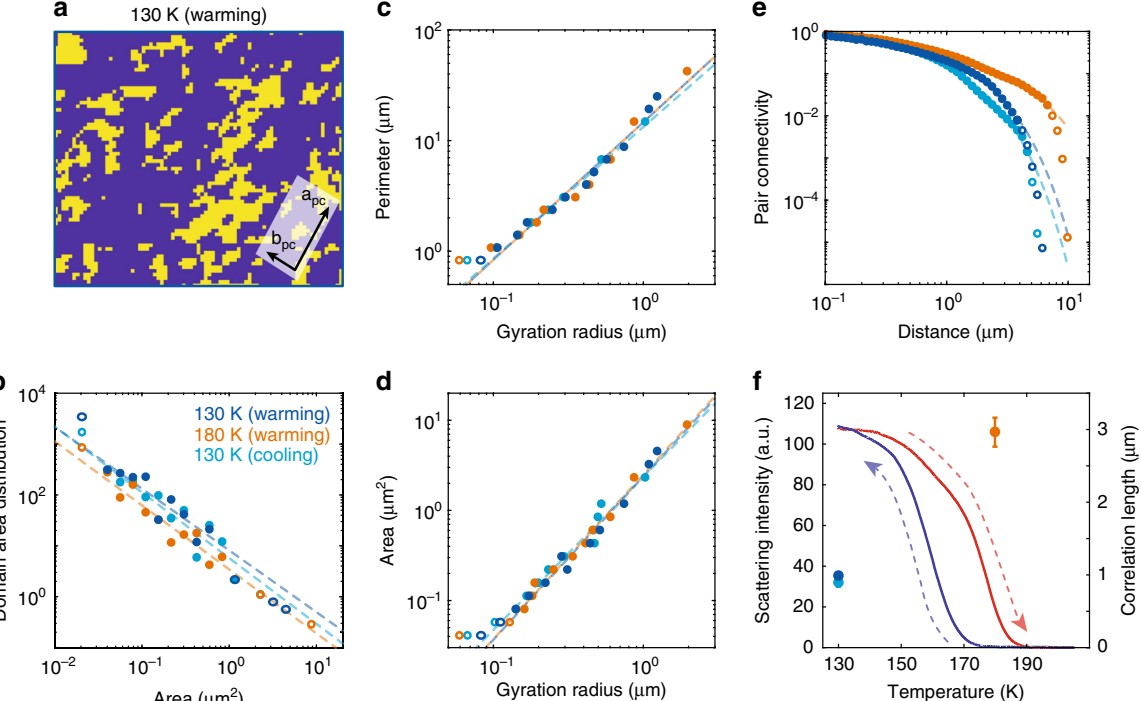

**Fig. 3** Fractal magnetic patterns. **a** The AFM domain map at 130 K (warming cycle) is binarized to highlight the AFM domains at the ordering vector probed here (yellow), vs. other, symmetry-equivalent ones to which we are not sensitive in the present geometry (blue). **b** The logarithmically binned AFM domain area distributions follow a scale-free power-law distribution ($D \sim A^{-\tau}$) with the critical exponent $\tau = 1.25 \pm 0.04$. Dashed lines are power-law fits to the experimental data points. Hollow markers represent points excluded from the fit. **c, d** Domain perimeter ($P$) and area ($A$) vs. gyration radius ($R_g$) with logarithmic binning. Dashed lines are power-law fits of $P \sim R_g^{d_h}$ and $A \sim R_g^{d_v}$ with the critical exponents $d_h = 1.23 \pm 0.03$ and $d_h = 1.78 \pm 0.07$. The power-law scaling and corresponding critical exponents ($d_h$, $d_v$) reveal a robust scale-invariant texture at all temperatures. **e** Pair connectivity function vs. distance ($r$) with logarithmic binning. The dash lines are fits to a power-law function with an exponential cutoff $g_{conn} \sim r^{-\eta} e^{-x/\xi}$ where $\xi$ is the correlation length and $\eta = 0.32 \pm 0.13$ is the exponent for the connectivity function. **f** Overlay of the temperature dependence of the $[\frac{1}{4}, \frac{1}{4}, \frac{1}{4}]_{pc}$ AFM Bragg peak intensity with the correlation lengths extracted from the pair connectivity function. The error bars for correlation lengths at 130 K are smaller than marker size

the correlation length, expected to diverge near criticality. The fitting result indicates that the correlation length at 180 K is the largest among the three maps. Moreover, the scaling exponent of the pair connectivity function is also far from the uncorrelated disorder model (Supplementary Table 1). Figure 3f shows the $\xi(T)$ data points overlaid onto the temperature dependence of the AFM Bragg peak intensity. The correlation length exhibits a clear enhancement (3×) near $T_{\text{Néel}}$, as expected in the critical theory.

To explain the observed scaling behavior, we note that when the system undergoes a phase transition near a critical point, its fluctuations become long-ranged, and multiple physical properties display power-law scaling with characteristic critical exponents[27,28]. Here, owing to domain pinning, potentially arising from quenched disorder, we can recover and visualize scale-invariant spatial correlations even down to 130 K, well below the critical temperature. The power-law scaling in the domain area distribution, the hull fractal dimension, the volume fractal dimension, and the pair connectivity function are all indicators of near-critical behavior, which has been previously discussed in the case of rare earth nickelates[48–50]. Moreover, these critical exponents satisfies the hyperscaling relation $d - 2 + \eta = 2$ $(d - d_v)$, corroborating the validity of our analysis[45]. Furthermore, while the correlation length diverges at a critical point, it is expected to decrease away from a critical point as $\xi \sim |T - T_{\text{Néel}}|^{\nu}$, providing a cutoff to the power-law scaling. This effect is evident in the 130 K cooling and 130 K warming datasets, where the correlation length of the pair connectivity function is about three times smaller than it is in the 180 K warming case (Fig. 3f).

## Discussion

The present results shed light on the complex mechanisms underlying the electronic and magnetic phase transition in nickelate compound NdNiO$_3$, conclusively demonstrating the near-critical nature of the AFM ordering transition. The direct visualization of magnetic domains presented here provides key complementary information to a series of previous studies of the nanoscale organization of the charge degrees of freedom, completing a body of experimental evidence from which we can now conclude that, in the rare earth nickelates: (i) the electronic domains form by nucleation and growth across a hysteretic, first-order IMT[7,11]; (ii) the magnetic domains are subordinated to the insulating regions within which they form, thus forcing the global order parameter to exhibit hysteresis (Fig. 2d); (iii) the AFM domains manifest the spatial signatures of scale invariance and critical behavior (Fig. 3), here rendered static by the pinning action of quenched disorder (the same pinning potential that likely underlies the observed memory effect); (iv) the critical behavior observed here is purely associated to the spin degree of freedom, which points to a different type of universality class as the continuous transition reported in ref. [10], which is only defined in spatial regions of vanishing magnetic order.

The methods and outcomes of the present study reach beyond the platform of rare earth nickelates. On the verge of criticality, the close interplay between competing phases triggers new organizing principles, with the underlying electronic fabric exhibiting spatial textures across all length scales. We have shown that nano-RMXS is a powerful imaging technique to visualize

multiscale electronic textures in complex oxides and diagnose critical behavior from a different perspective, and one that is complementary to temperature scaling analysis. In this context, we find unexpected scale-invariant AFM domain patterns in $NdNiO_3$ thin films, which directly demonstrate the continuous nature of the magnetic phase transition in this system and extend the paradigm of nanoscale phase inhomogeneity to correlated electron systems with an AFM ground state. Together with prior studies, the reported observations reaffirm the critical need for nanoscale probes of local order parameters to examine electronic symmetry-breaking phenomena, whose true nature may evade conventional approaches based on macroscopic measurements. These methods have important implications for the experimental protocols aimed at assessing the nature and symmetry of electronic phase transitions in the presence of inhomogeneity and coupled degrees of freedom. In light of these aspects, the present work paves the way for future explorations of emergent electronic and magnetic phases in strongly correlated quantum materials near criticality.

## Methods

**Samples**. The $NdNiO_3$ $(001)_{pc}$ film was grown epitaxially on a commercial $NdGaO_3$ (110) substrate by off-axis radio-frequency magnetron sputtering. The film showed a hysteresis loop in the metal-to-insulator transition (MIT) as shown in Supplementary Fig. 1a. The MIT temperature is 178.4 K and 168.7 K for the warming and cooling cycle respectively, determined from where $dR/dT$ changes sign. The inset shows an atomic force microscopy image of the atomic terraces at the film surface. The X-ray reflectivity (Supplementary Fig. 1b) measurements indicate a film thickness of 26 nm, corresponding to 69 unit cells, which is thick enough to exhibit bulk properties. Prior to the experiment, a semitransparent Cr fiducial mask was deposited onto the sample surface for position registration purpose during the scanning X-ray scattering measurements. The grid layout, as well as an optical microscope view of the Cr fiducial mask, are shown in Supplementary Fig. 1c and d. The thickness of the Cr grid line is 143 nm as measured by atomic force microscopy.

**Nano-RMXS**. The resonant soft X-ray scattering nanoprobe (nano-RMXS) experiment was performed at beamline CSX (23-ID-1) of the National Synchrotron Light Source II, where a bright and almost fully coherent soft X-ray photon beam is delivered at the sample. A 20 μm pinhole was used to magnify the X-ray beam to around 200 μm diameter at the zone plate location (1 m downstream of the pinhole). The focusing setup includes a gold Fresnel zone plate (FZP) and an order sorting aperture (OSA), mounted on two independent nanopositioning stages. The FZP has a diameter of 240 μm and a 95 μm central beam stop to block the $0^{th}$ order transmission. The FZP outer zone width is 35 nm, which gives a focal length of 5.7 mm for the first diffraction order, at the Ni-$L_3$ resonant edge (852 eV). The combination of FZP, central beam stop, and OSA, is used to select the first diffraction order.

**XPEEM**. XPEEM was measured at beamline ESM (21-ID-2) of the National Synchrotron Light Source II.

## Data availability

The data that support the plots within this paper and other findings of this study are available from the corresponding author upon reasonable request.

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

## Acknowledgements

The authors would like to thank B. Keimer, E. Benckiser, M. Bluschke, D. Basov, S. Ramanathan, G. Mattoni, A. Caviglia, G. Sawatzky, R. Green, P. Evans, and I. Schuller for insightful discussions. This material is based upon work supported by the National Science Foundation under Grant No. 1751739. This research used resources of the National Synchrotron Light Source II and the Center for Functional Nanomaterials, which are US Department of Energy (DOE) Office of Science facilities at Brookhaven National Laboratory, under Contract No. DE-SC0012704. This work was supported by the Swiss National Science Foundation through Division II. The research leading to these results has received funding from the European Research Council under the European Union's Seventh Framework Program (FP7/2007-2013)/ERC Grant Agreement no. 319286 (Q-MAC). E.W.C. and F.S. acknowledge support from NSF DMR-1508236, Depasrtment of Education Grant No. P116F140459, and the Purdue Research Foundation. J.P. acknowledges financial support by the Swiss National Science Foundation Early Postdoc Mobility fellowship project number P2FRP2_171824 and PostDoc Mobility project number P400P2_180744.

## Author contributions

J.L., C.M., and R.C. conceived the research study, especially with sample and setup design. S.C., M.G., and J.M.T. synthesized the samples, and carried out materials analysis and characterization. J.L., J.P., C.M., A.L., S.W., and R.C. contributed to the nano-RMXS experiment. J.L. and J.T.S carried out XPEEM measurements. J.L., F.S., E.C., and R.C. performed data analysis and interpretation. J.L. and R.C. wrote the manuscript, with input and edits from all co-authors.

## Competing interests

The authors declare no competing interests.
