## [Peer Review File · Nature Communications]

Reviewers' comments:

Reviewer #1 (Remarks to the Author):

The work of Li et al reports on the application of resonant-X-ray-based magnetic imaging with 100 nm lateral spatial resolution in thick NdNiO₃ films. AFM domains are imaged across the Neel transition in this material, to elucidate the nanoscale magnetic texture that develops in such systems. As the authors well summarize in their introduction, there are many open questions in this area in nickelates. What the authors find is a scale invariant AFM domain structure which returns to a similar state after warming through the transition temperature and re-cooling. This evidences (clearly) strong pinning of the domain walls to underlying structural/chemical inhomogeneities. The authors then present a detailed analysis of the images, concluding proximity to a critical point, with quenched disorder playing an important role. In the paragraph beginning "The present results..." the authors then nicely put their work in context with the recent literature findings and current thinking.

There is no doubt this is a good contribution to the literature in this area. The method is non-trivial, and it is applied to good samples. The finding of strong pinning of AFM domain walls is not so surprising, but the ensuing scale analysis is informative. Altogether, this clearly adds something to the understanding in the field. I am just not 100% certain that this reaches the level the editors are looking for in Nature Communications. I have only one technical question for the authors. The term "return point memory" is frequently used in the description of hysteresis vs magnetic field. Is the term also in common usage for thermal hysteresis, as they use it?

In summary, I think this is strong work, but I am not 100% convinced I see a single innovation or breakthrough I would consider as sufficient for this highly selective journal.

Reviewer #2 (Remarks to the Author):

The authors present a novel application of the resonant magnetic X-ray scattering technique to reveal antiferro magnetic domains in nickelate films to elucidate the nature of the second-order phase transition in the system. The detection of AF domains at these length-scales provides a wealth of knowledge with regards to the understanding of critical behavior and will be of wide interest to the community. A minor comment I have will be that the authors comment on the depth-dependence of the domains. The measurements are clearly presented and analysed and I support the publication of this manuscript in Nature communications.

Reviewer #3 (Remarks to the Author):

The authors report on a scale-invariant magnetic texture in a strongly correlated oxide, studied by means of soft X-ray nano diffraction.

The main conclusions of the manuscript are that

- i) the magnetic domains are subordinated to the insulating regions within which they form, thus forcing the global order parameter to exhibit hysteresis
- ii) the AFM domains manifest the spatial signatures of scale invariance and critical behavior demonstrating the critical nature of the AFM transition.
- iii) it can be argued that the spatial regions (conductive 1D channels) previously found to order continuously across the first-order IMT are indeed the boundaries of AFM domains

Looking at the criteria for publication in NC I believe that the manuscript presents data which is novel and it is technically sound

and provides strong evidence for its conclusions, except maybe for point iii).

The manuscript will definitely be important for scientists in the specific field and as such warrants

publication in some form.

What I'm not really sure about is whether the work represents an advance that is "likely to influence thinking in the field".

Soft x-ray nanodiffraction studies of antiferromagnetic domains have been presented already in Phys. Rev. B 88, 075134 (2013) and therefore what is new

here is the application of statistical methods to the AFM domains to infer critical properties of the phase transition.

In this respect, I believe that these methods are known by the strongly correlated electron system community as they have been highlighted already

in publication of one of the co-authors in the last years (e.g. NC 1920 (2012)).

Regarding conclusions i) and ii), it is very nice and important to have a direct proof of such behaviour, however I feel that the result here does not really come as a big surprise for those in the field.

Regarding point iii), I would have found it very interesting if the manuscript could have provided a more convincing evidence of different/same

critical behaviour of the phase transition in the 1D domain boundaries as suggested by Ref. 10 of the manuscript.

In the domain boundaries the transition is claimed to be of 2nd order, and based on the analogy with another member of the RNiO₃ family one could expect

that the AFM ordering would develop at lower temperature as compared to the IMT.

Is there anything in the data gathered during the experiment that could be used to prove/disprove this scenario?

This would really be something of general interest and therefore would make sure the manuscript fulfils all the publication criteria.

Focusing on the manuscript, I found it in general well written. However, I recommend to be less technical at some specific points,

to help the reader follow the flow, especially if the authors are aiming at publishing in a non-specialized journal.

line

xxxx

Comments and Questions:

1) Drift-corrected intensity map from A. How the correction is performed?

2) How is the scanning performed? Is there one or more alignment scan at each temperature, each position?

3) Correlation length of the diffraction peak in the Temperature scan: how does it compare with the one inferred from their modelling?

4) How does it compare with the neutron results reported by Europhys. Lett. 20, pp. 241-247 (1992) ?

5) discussion in lines 171-176: this discussion is difficult to follow. No reference is given for the cited percolation critical exponents.

It is cited that "interactions play a crucial role", but this statement is very vague. Are we talking about Coulomb interaction, exchange interaction, etc...

line 55: I would add a reference after (Nd and Pr).

line 127: please define what you mean by a "stellated" octahedron.

line 128-130: It will be helpful to specify if the magnetic structure factor for some of the domains is zero because of the azimuthal angle setting in the experiment or not.

line 161: Please provide more detail on how the logarithmic binning is performed.

line 170: How is r defined and how the R_g is calculated? What are the uncertainties in the value of d_h and d_v ?

line 179: what is meant by "non-compact"?

We would like to thank all reviewers for taking the time to examine and assess our manuscript, and for the valuable comments and remarks that have guided our revisions and inspired further analysis of the reported experimental results.

Below we enclose a point-by-point response to all reports, to which follows a 1-page addendum with a summary of all changes to the manuscript (main + SI). The changes in the revised documents have been highlighted in red font.

Yours sincerely,

Riccardo Comin, on behalf of all authors

Reviewer #1 (Remarks to the Author):

The work of Li et al reports on the application of resonant-X-ray-based magnetic imaging with 100 nm lateral spatial resolution in thick NdNiO₃ films. AFM domains are imaged across the Neel transition in this material, to elucidate the nanoscale magnetic texture that develops in such systems. As the authors well summarize in their introduction, there are many open questions in this area in nickelates. What the authors find is a scale invariant AFM domain structure which returns to a similar state after warming through the transition temperature and re-cooling. This evidences (clearly) strong pinning of the domain walls to underlying structural/chemical inhomogeneities. The authors then present a detailed analysis of the images, concluding proximity to a critical point, with quenched disorder playing an important role. In the paragraph beginning “The present results...” the authors then nicely put their work in context with the recent literature findings and current thinking.

There is no doubt this is a good contribution to the literature in this area. The method is non-trivial, and it is applied to good samples. The finding of strong pinning of AFM domain walls is not so surprising, but the ensuing scale analysis is informative. Altogether, this clearly adds something to the understanding in the field. I am just not 100% certain that this reaches the level the editors are looking for in Nature Communications. I have only one technical question for the authors. The term “return point memory” is frequently used in the description of hysteresis vs magnetic field. Is the term also in common usage for thermal hysteresis, as they use it?

In summary, I think this is strong work, but I am not 100% convinced I see a single innovation or breakthrough I would consider as sufficient for this highly selective journal.

Our response: We would like to thank Reviewer 1 for taking the time to assess out manuscript, and for the appraisal of our study. We hope the Reviewer might concede us an opportunity to clarify the main elements of novelty of our study, and to articulate the reasons why our manuscript is a good fit for *Nature Communications*.

The methods and materials that are central to our study are contiguous to those of two recent manuscripts published in this journal: Mattoni et al. Nat. Comm. 7, 13141 (2016) (imaging of charge domains in nickelates) and Kim et al Nat. Comm. 9, 5013 (2018) (Bragg topographic imaging of antiferromagnetic antiphase domain boundaries). The former paper focused on revealing new aspects of the electronic phase separation in nickelates, through the analysis of spatial textures and nanoscale organization of electronic domains. The latter described the development of a new method for imaging bulk antiferromagnetic domains in complex oxides, albeit at greater length scales and fields of view compared to our study.

Our manuscript is a combination of both elements: (a) the development of scanning resonant nanodiffraction at sub-100 nm length scales – an unprecedented and significant experimental step forward to observe these multiscale spatial textures [with resolution outperforming both the studies of Nat. Comm. 9, 5013 (2018) and PRB 88, 075134 (2013)]; and (b) the analysis and discovery of a scale-invariant antiferromagnetic domain structure and its significance as a hallmark of critical behavior of the magnetic degrees of freedom, addressing an important open question that was raised but not conclusively answered in a recent study (Nat Phys 14, 1056 (2018)), as also remarked by reviewer 3. The fusion of these aspects enabled us to significantly advance our description and understanding of the nature of the magnetic transition in nickelates, including its critical exponents (this has been an open question since

these materials were originally discovered) while at the same time developing methods with broad application to visualize electronic domains in many other materials. We would like to emphasize that these are the first nanoscale maps of antiferromagnetic domains ever reported in the RENiO₃ family, and that these new data played a crucial role in revealing the critical nature of the AFM transition in this class of compounds – an outstanding problem that could not be addressed by other techniques thus far.

In our view, all the above reasons made, and still make, our manuscript and study an excellent fit to Nature Communications and its readership.

Regarding the matter of the use of the term “return point memory”, we would like to note that it is generally accepted in reference to any form of hysteresis, regardless of the type of control variable. However, as the referee points out, it is less familiar when the control variable is temperature. Therefore, we have changed “return point memory effect” into “memory effect,” so as not to distract the reader with terminology.

Reviewer #2 (Remarks to the Author):

The authors present a novel application of the resonant magnetic X-ray scattering technique to reveal antiferromagnetic domains in nickelate films to elucidate the nature of the second-order phase transition in the system. The detection of AF domains at these length-scales provides a wealth of knowledge with regards to the understanding of critical behavior and will be of wide interest to the community. A minor comment I have will be that the authors comment on the depth-dependence of the domains. The measurements are clearly presented and analysed and I support the publication of this manuscript in Nature communications.

Our response: We would like to thank Reviewer 2 for taking the time to assess our manuscript, and for the positive feedback on the significance of our study. Regarding the depth-dependence of the domain structure, this is an excellent point: the thickness of the sample is 26 nm which is lower than the average cluster size hovering around few hundreds of nm. Therefore, these measurements are performed on a sample that is close to the 2D limit for what regards the organization of the magnetic ground state and its domain structure.

Reviewer #3 (Remarks to the Author):

The authors report on a scale-invariant magnetic texture in a strongly correlated oxide, studied by means of soft X-ray nano diffraction. The main conclusions of the manuscript are that:

- i) the magnetic domains are subordinated to the insulating regions within which they form, thus forcing the global order parameter to exhibit hysteresis;
- ii) the AFM domains manifest the spatial signatures of scale invariance and critical behavior demonstrating the critical nature of the AFM transition;
- iii) it can be argued that the spatial regions (conductive 1D channels) previously found to order continuously across the first-order IMT are indeed the boundaries of AFM domains

Our response: We thank Reviewer 3 for taking the time to examine closely and thoroughly our manuscript, and for discussing the very relevant context to prior studies in the enclosed remarks. While the Reviewer correctly pointed out the various central elements of the present study, we would like to start our response with a brief note on (iii): while it is correct that the technique of ref 10 is potentially sensitive to boundaries of AFM domains, these were proposed to be contours where the magnetic order parameter goes to zero (creating a conducting channel), therefore antiphase boundaries between domains with the same wavevector Q . Our measurements instead exclusively map out regions exhibiting magnetic order with a given ordering vector Q and are, therefore, insensitive to the type of antiphase domain boundaries discussed in ref 10. It is also worth noting that the domain boundaries visualized in ref 10 are sensed through their higher conductivity, whereas our method is a direct measurement of the local AFM order parameter. In light of this new analysis, we have updated the discussion of the relationship between our findings and the results reported in ref. 10. This revised part was correspondingly reworded to highlight the major advancements afforded by our method of measurement (direct imaging of magnetic domains) and the uniqueness of our conclusions on the nature of magnetic criticality in this system.

Looking at the criteria for publication in NC I believe that the manuscript presents data which is novel and it is technically sound and provides strong evidence for its conclusions, except maybe for point iii). The manuscript will definitely be important for scientists in the specific field and as such warrants publication in some form. What I'm not really sure about is whether the work represents an advance that is "likely to influence thinking in the field".

Soft x-ray nanodiffraction studies of antiferromagnetic domains have been presented already in Phys. Rev. B 88, 075134 (2013) and therefore what is new here is the application of statistical methods to the AFM domains to infer critical properties of the phase transition. In this respect, I believe that these methods are known by the strongly correlated electron system community as they have been highlighted already in publication of one of the co-authors in the last years (e.g. NC 1920 (2012)). Regarding conclusions i) and ii), it is very nice and important to have a direct proof of such behaviour, however I feel that the result here does not really come as a big surprise for those in the field.

Our response: We would like to thank the Reviewer for raising these specific remarks (i)-(iii) about novelty of our work and for evaluating the significance our results within the broader state of the field. These remarks pinpoint various elements of this study that were, in fact, not discussed with sufficient context and emphasis. We would like to first address remarks (i)-(ii) and later discuss point (iii):

- (i) the Reviewer is correct that the presence of hysteresis in the global (spatially-averaged) order parameter is known, both from previous studies (Ref. 38 in the maintext) and from data reported in the present manuscript (Fig. 2). This fact is directly derived from the temperature dependence of the overall magnetic scattering intensity and, despite being reported here, is not one of the *main conclusions* of our work – this behavior was in fact well-expected, also from existing literature. What the prior studies could not elucidate, until now, is how this hysteresis comes about. The phenomenological presence of hysteresis could be due to two mechanisms: (a) magnetic domains forming according to a first-order phase transition, namely by nucleation and growth, with the two cycles exhibiting different nucleation dynamics causing an overall hysteresis; or (b) a magnetic order parameter developing in a continuous fashion, but constrained within (insulating) domains which themselves behave hysteretically, in line with the first-order nature of the metal-insulator transition.

In the present study, we have determined that scenario (b) is the one at play, as is experimentally evident from the comparison between the maps at 130 and 180 K: the evolution observed in the warming cycle, leading to a suppression of a factor 6 of the global order parameter is not due to shrinking magnetic domains (as would be the case for scenario (a) above) but rather to a reduction in the order parameter *within* each domain. This local reduction is of the same order as that in the spatially-averaged order parameter -- suggesting that, while the domain structure remains largely unchanged, the order parameter is *uniformly* suppressed across the transition. This mechanism is only compatible with scenario (b), and provides complementary evidence of near-critical behavior, as also determined from the analysis of the spatial domain textures. This discussion has now been included in the revised manuscript to provide further support that the magnetic transition is *continuous*, rather than driven by domain nucleation and growth.

- (ii) we would like to argue that the presence of signatures of scale invariance and static power law domain distribution is, in fact, quite different from what has been discussed about magnetic transitions prior to this study. Typically, the formation of static magnetic domains is discussed in the context of a first order phase transition, which transpires by nucleation and growth of independent domains which then solidify. When a second order transition is discussed in regards to Néel order parameters, critical fluctuations naturally transpire near the Néel transition temperature, but in a very different way from what is observed here. Critical fluctuations in that scenario are dynamic in time, and do not produce the static domains we observe. Because the magnetic domains observed in this study are both power-law distributed and static in time, that points to disorder-driven criticality of the random field type. Random field disorder is any disorder which couples linearly to the order parameter. It is a particularly severe class of disorder, in that it renders temperature fluctuations irrelevant in the renormalization group sense (G. Tarjus et al, EPL 103, 61001 (2013)). A consequence of this fact is that barriers to equilibration skyrocket, and the dominant domain fluctuations are in space rather than in time – that is to say, the domains become glassy as we reported in this work. In short, while static domains are familiar from first order phase transitions, and power law behavior is familiar from second order phase transitions, finding domains which are both power law and static means the criticality is driven by random-field disorder. If only random fields were at play in the absence of interactions, the universality class would be that of uncorrelated percolation. Therefore, the strong shift of these exponents away from percolation also indicates that interactions between the magnetic order parameter from one region to the next are also important. These are not trivial or generally expected findings in magnetic systems.

We have found that the previous version of the manuscript was not clear on the above aspects. We have correspondingly reworded and expanded the discussion of page 3 to convey these important aspects that make our observations quite special in the context of antiferromagnetic transitions.

Regarding point iii), I would have found it very interesting if the manuscript could have provided a more convincing evidence of different/same critical behaviour of the phase transition in the 1D domain boundaries as suggested by Ref. 10 of the manuscript. In the domain boundaries the transition is claimed to be of 2nd order, and based on the analogy with another member of the RNiO₃ family one could expect that the AFM ordering would develop at lower temperature as compared to the IMT. Is there anything in the data gathered during the experiment that could be used to prove/disprove this scenario?

This would really be something of general interest and therefore would make sure the manuscript fulfils all the publication criteria.

Our response: We would like to thank the referee for encouraging us to go deeper in elucidating the relation of our work to Ref. 10. The referee has raised some very stimulating ideas in this regard, as discussed below.

- Concerning the idea of “different/same critical behavior”:

We have re-examined our understanding of the link between our results and the IR nanospectroscopy measurements of Ref. 10 (Post *et al.*). The measurements of Ref. 10 report a continuous evolution of a metallic order parameter which develops on what is proposed to be antiphase lines in the magnetic order parameter. The metallicity is confined to those regions because they are envisioned to be the only places where the magnetic order parameter vanishes. Thus, when discussing the Landau-Ginzburg theory of the metallic channels, Ref. 10 implicitly assumes that the magnetic order parameter is locally well-developed everywhere else, i.e. that the magnetic order parameter is not close to criticality in the temperature regime in which the metallic channels are observed.

Conversely, our nano-RMXS measurements directly image the AFM domain patterns, and our probe is selectively sensitive to the magnetic order parameter. The local metallicity is not detected in the present measurements, which implies that the power law scaling observed here is to be ascribed exclusively to the proximity to a magnetic critical point.

In the spirit of this new analysis, we can firmly conclude that the criticality of the metallic channels is different from the magnetic criticality we have identified in this manuscript, not only because the symmetries of the two order parameters are different, but also because they are occurring at different temperatures. What this study can rigorously affirm is that below the Neel transition temperature, there is spatial complexity in the domain structure in the form of power law scaling. This is strong evidence in favor of one of the assumptions of Ref. 10, that on the length scales of their measurement, magnetic domains are both well defined and also have significant structure.

- Concerning the idea that “one could expect that the AFM ordering would develop at a lower temperature as compared to the IMT”:

Because Ref. 10 interpreted the metallic channels as arising from kinks in the magnetic order parameter, that includes the implicit assumption that T_{Neel} is higher than the temperature regime in which the metallic channels form. Unfortunately, to really nail this question empirically of whether, as the referee points out, the AFM could rather be developing at a lower temperature than the IMT, that would require in situ measurements of the charge and spin degrees of freedom simultaneously, which goes beyond the present dataset.

- Concerning the matter of general interest: we note that in both the Post *et al* and this work, while the macroscopic measurements point to first order transitions in both charge and spin degrees of freedom, the nanoscale measurements tell a different story: It has now been shown that not only is there a second order transition in the metallic degrees of freedom at the nanoscale, but that there is also a second order transition in the magnetic degrees of freedom at the nanoscale. We expect this dichotomy between the evidence provided by macroscopic and nanoscale measurements, which we have now shown to be a more general feature of these materials, to be of broad interest. Taken together, these nanoscale measurements challenge the standard symmetry classifications of phase transition theory, which are based on macroscopic definitions and macroscopic criteria.

We have reworded the content of the last two paragraphs of the main text to report the outcome of the analysis undertaken in response to this remark, and to convey these concepts more clearly.

Focusing on the manuscript, I found it in general well written. However, I recommend to be less technical at some specific points, to help the reader follow the flow, especially if the authors are aiming at publishing in a non-specialized journal.

Comments and Questions:

1) Drift-corrected intensity map from A. How the correction is performed?

Our response: The drift correction process is described in the *Data acquisition* section of the supplementary information. More specifically, the correction was performed by using a fiducial Cr grid as a reference frame for the position drift. The drift-corrected intensity map was generated via a local affine transformation guided by the known shape of the Cr grid.

2) How is the scanning performed? Is there one or more alignment scan at each temperature, each position?

Our response: Before acquiring each spatial map at different temperatures shown in the manuscript, the Cr mask was scanned to ensure all measurements are performed in the same field of view and the drift is minimal. This is, in fact, an important detail that has already been described in the *Data acquisition* section of the supplementary information.

3) Correlation length of the diffraction peak in the Temperature scan: how does it compare with the one inferred from their modelling?

Our response: This is an excellent question, on a matter that is quite technical, and deserved to be mentioned in the text (it is now added to the SI). The macroscopic beam measurement is performed by moving the sample away from the focus of the zone plates, where the beam footprint is approximately 200 μm in diameter. While the spot is large, the photon beam remains strongly divergent (by virtue of passing through the zone plate focusing element), consequently the width of the diffraction peak is largely determined by the divergence of the X-ray beam, with the contribution from the intrinsic spatial correlations being minor. As a result, the AFM correlation lengths could not be deconvolved from the diffraction peak linewidth.

4) How does it compare with the neutron results reported by Europhys. Lett. 20, pp. 241-247 (1992) ?

Our response: Our data from spatially-averaged resonant magnetic X-ray diffraction agree well with the neutron results of García-Muñoz, showing similar signatures of first order transition as also discussed above in the context of remark (i).

5) discussion in lines 171-176: this discussion is difficult to follow. No reference is given for the cited percolation critical exponents.

Our response: Thanks for pointing out the difficult language of this passage. The sentence that followed, in the original version, already included an appropriate reference but we have additionally reworded the text to be clearer and more accessible to the reader.

It is cited that "interactions play a crucial role", but this statement is very vague. Are we talking about Coulomb interaction, exchange interaction, etc...

Our response: We mean that the magnetic order parameter in one region interacts with the magnetic order parameter in neighboring regions. The origin of this interaction (as well as that of the exchange interaction) is ultimately Coulombic in strongly correlated systems.

line 55: I would add a reference after (Nd and Pr).

Our response: The reference has been added in the text.

line 127: please define what you mean by a "stellated" octahedron.

Our response: Thanks for pointing out the unclear definition. We have removed the misleading "stellated octahedron" and expanded the discussion of this aspect with more accessible words.

line 128-130: It will be helpful to specify if the magnetic structure factor for some of the domains is zero because of the azimuthal angle setting in the experiment or not.

Our response: Yes, this is correct. This aspect was originally mentioned in the footnote of page 5 and has now been moved to the main text to make this detail more conspicuous.

line 161: Please provide more detail on how the logarithmic binning is performed.

Our response: For the case of logarithmic binning, the bin sizes are constructed to maintain a constant ratio between consecutive terms. The first bin contains the smallest domains while the last bin contains the largest domain. Ratio between successive bins is select to be 1.4, 1.3 and 1.4 for Fig. 3b-d, respectively, to ensure a fair sampling of domains while ensuring a large enough number of bins to fit the exponents.

line 170: How is r defined and how the R_g is calculated? What are the uncertainties in the value of d_h and d_v ?

Our response: For each pixel, the position of the X-ray spot was projected onto the coordinate system of the sample frame (as shown in Fig. 2e). The position vector r is defined in the sample coordinate system. The gyration radius for each individual domain $R_g = \sqrt{\langle (r - \langle r \rangle)^2 \rangle}$ is calculated as the root mean square distance of all pixels in the domain from its center of mass. The estimates and uncertainties for d_h and d_v are reported in Figs. 3c,d as $d_h = 1.23 \pm 0.03$, $d_v = 1.78 \pm 0.07$.

line 179: what is meant by "non-compact"?

Our response: Non-compact means that the interior dimension of the clusters is a non-integer number.

SUMMARY OF CHANGES

In this document, we summarized the revisions of the main manuscript and Supplementary Information file of the article “Scale-invariant magnetic textures in a strongly correlated oxide”. Edits are highlighted in red in the revised documents. When applicable, we also reference the reviewer’s comment addressed by a given item. Square brackets enclose the beginning of each section and line numbers:

- 1) **[Authors: ...]** – We updated the affiliation for one of the authors in line 4 and line 12.
- 2) **[The nature of the AFM transition ... (line 54-55)]** – We have added references as suggested by Reviewer 3.
- 3) **[In the present work ... (line 76)]** – We have expanded this paragraph to provide more extended background information on the theory of static critical behavior of magnetic order in NdNiO₃ in the presence of random-field disorder.
- 4) **[The presence of AFM spatial textures ... (line 126)]** – We have reworded this paragraph to clarify the symmetry relationship between AFM ordering vectors, as suggested by Reviewer 3.
- 5) **[As the sample is cooled back down ... (line 146)]** – We have changed the term “return point memory effect” to “memory effect” as suggested by Reviewer 3.
- 6) **[The temperature-dependent spatial maps ... (line 156)]** – This is a new paragraph, introduced to provide further information on the continuous nature of the AFM transition in NdNiO₃, in response to Reviewer 3.
- 7) **[The availability of ... (line 172 - 205)]** – This paragraph is unchanged except for minor rephrasing in response to the comments of Reviewer 3.
- 8) **[To explain the observed scaling behavior, ... (line 217-230)]** – This paragraph is unchanged except for a minor rephrasing.
- 9) **[The present results ... (line 232 - 248)]** – We have revised conclusion (iv) in consideration of the new analysis of antiphase domain walls vs. bulk domain patterns, inspired by Reviewer 3.
- 10) **[The methods and outcomes of the present study ... (line 249 - 262)]** – We have rephrased this paragraph to emphasize the novelty and implications of our study in relation to the present status and understanding of these materials and scientific questions.
- 11) **[References]** – We have updated the references accordingly.

No changes have been made to figures and captions in the main text. In the Supplementary Information, we made following changes in each section:

- 1) **[The macroscopic beam measurement ... (line 45 - 54)]** – We have included additional details on the macroscopic beam measurements to address one of the questions from Reviewer 3.
- 2) **[Memory effect (line 80)]** – We have changed the term “return point memory effect” to “memory effect” as pointed out by Reviewer 3.
- 3) **[Signatures of continuously-evolving magnetic order parameter (line 99 - 107)]** – This is a new section focusing on the analysis of the magnetic scattering intensity inside the AFM domains. The new analysis clearly demonstrated that the magnitude of magnetic order parameter is being uniformly suppressed as a function of temperature rather than being controlled by mechanisms of domain growth and nucleation.
- 4) **[Supplementary Figures]** – We have added Fig. S6 in line with the section **Signatures of continuously-evolving magnetic order parameter**. The remaining SI figures are unchanged.

REVIEWERS' COMMENTS:

Reviewer #1 (Remarks to the Author):

I have read the revised paper, the comments of all referees, and the responses to those comments. I have to say that my opinion of the likely impact of this work has improved. The authors responses to both my own comments and questions, and those of referee 3, are convincing and comprehensive. The new text added in response to these comments also significantly improves the readability and accessibility of the arguments made. Most importantly, I think I now have a more positive impression of the novelty of this work with respect to prior studies, and how these observations differ from the norm. I would now support publication in Nature Communications. In particular, the manner in which the authors are able to place their new findings in the context of prior work to provide a higher level view of these specific nickelates is impressive.